# A tipping point in stable isotope composition of Antarctic meteoric waters during Cenozoic glaciation

**Luigi Dallai** [1,2] ✉ **& Zachary D. Sharp**[3,4]

Triple oxygen isotopes of Cenozoic intrusive rocks emplaced along the Ross Sea coastline in Antarctica, reveal that meteoric-hydrothermal waters imprinted their stable isotope composition on mineral phases, leaving a clear record of oxygen and hydrogen isotope variations during the establishment of the polar cap. Calculated O- and H-isotope compositions of meteoric waters vary from −9 ± 2‰ and −92 ± 5‰ at 40 ± 0.6 Ma, to −30 and −234 ± 5‰ at 34 ± 1.9 Ma, and intersect the modern Global Meteoric Water Line. These isotopic variations likely depict the combined variations in temperature, humidity, and moisture source regions, resulting from rearrangement of oceanic currents and atmospheric cooling during the onset of continental ice cap. Here, we report a paleo-climatic proxy based on triple oxygen geochemistry of crystalline rocks that reveals changes in the hydrological cycle. We discuss the magnitude of temperature changes at high latitudes during the Eocene-Oligocene climatic transition.

Variations in the continental hydrological cycle during Earth's evolution are crucial to our understanding of atmospheric conditions in the geological past and defining boundary conditions for future temperature projections. Changes in the water cycle, such as changes to atmospheric and oceanic circulation, variations in atmospheric humidity, and changes in moisture transport, all contributed to reaching tipping points in the paleo-climate record.

During the Cenozoic, a prolonged trend of global temperature decrease brought the climatic system from Palaeocene/Eocene hyperthermals to Eocene/Oligocene glacial conditions[1]. This finding is based on many proxies provided by materials that equilibrated with ambient (meteoric/oceanic) waters, such as marine sediments, biological materials, and continental carbonates[2–4]. The sedimentary records from terrestrial margins of Antarctica have provided evidence for the expansion and retreat of land-terminating glaciers, which delivered eroded material to the sea, and indicate that stable, continental-scale ice sheets were established after ~32.8 Ma[5]. Whether the cooling that led to glacial conditions was forced by $p$CO$_2$ (partial pressure of CO$_2$ in the atmosphere) drawdown alone, or by

geodynamic, tectonic, and atmospheric mechanisms acting in concert, is a matter of debate[6–8]. A strong coupling between sea surface temperature (SST), mean air temperatures (MAT), and atmospheric $p$CO$_2$ has been established for the Eocene-Oligocene epochs[9–12]. Numerical simulations and geochemical records indicate that the drawdown of atmospheric CO$_2$ concentration[5,12] during this period led to the onset of ice caps, initially in continental interiors. Ice tongues subsequently expanded towards the continental margins, affecting continental and oceanic water balance (groundwater availability and evaporation), and ocean surface temperatures[13,14]. Climate models indicate that the effects of declining atmospheric $p$CO$_2$ concentration grew progressively[3,10,15–17], and co-vary with the records of cooling circum-Antarctic SSTs in the late Eocene[3,18].

Reduction of atmospheric $p$CO$_2$ influenced both atmospheric temperatures and precipitation, with the result of decreasing relative humidity and moisture availability[19]. Therefore, the stable isotope composition of Eocene meteoric waters should have varied in response to differences in the global hydrological cycle. Due to the meteoric origin of hydrothermal waters, the stable isotope

[1]Dip. Scienze della Terra, Università degli Studi di Roma "Sapienza", Roma, Italy. [2]CNR – IGG, Area della Ricerca di Pisa, Pisa, Italy. [3]Department of Earth and Planetary Sciences, University of New Mexico, Albuquerque, NM, USA. [4]Center for Stable Isotopes, University of New Mexico, Albuquerque, NM, USA. ✉ e-mail: luigi.dallai@uniroma1.it

compositions of rocks that interacted with hydrothermal waters can be used as a proxy for the meteoric water values responsible for the water/rock interaction[20–25].

In this work, we used the triple oxygen isotope ($^{17}O/^{16}O$, $^{18}O/^{16}O$) and hydrogen isotope geochemistry (D/H) of Cenozoic rocks belonging to local hydrothermal systems along the Ross Sea coastline in Antarctica, to estimate the initial oxygen and hydrogen isotope compositions of meteoric waters responsible for the hydrothermal alteration. We analysed alkaline intrusive rocks (diorite to qz-syenite and their country rocks) that pre-date and post-date the onset of the Antarctic glaciation, to constrain the isotopic composition of palaeo-meteoric waters and the prevailing atmospheric conditions at the Eocene-Oligocene (E-O) boundary.

## Results

### Interaction of meteoric waters with intrusive rocks

Alkaline intrusive rocks were emplaced along the Ross Sea Embayment in Antarctica from $51.6 \pm 0.6$ to $29 \pm 1.7$ Ma (Ar-ages)[25,26]. Plutonic bodies consist of amphibole (+/- biotite)-bearing syenites and monzonites, with subordinate monzogabbro/monzodioritic facies. During cooling, meteoric-hydrothermal waters circulated through the plutons and their country rocks producing low-to-intermediate temperature (250–350 °C) alteration. The lowest oxygen and hydrogen isotope values occur near the borders of the plutons due to higher water/rock (W/R) ratios where the magmatic bodies emplaced along crustal fractures[25]. Hydrothermal fluids are dominated by meteoric waters: their hydrogen isotopic composition is nearly identical to that of the local meteoric water, while their oxygen isotope composition is usually shifted to more positive δ-values due to water/rock interaction[27]. One can therefore use the stable isotope composition of hydrothermally altered rocks to estimate the meteoric water isotopic composition at the time of hydrothermal water/rock interaction.

The interaction between water and rock during hydrothermal alteration can be modelled in terms of simple mass balance mixing processes[28], and triple oxygen geochemistry has expanded the possibility to reconstruct the composition of the interacting water from hydrothermally altered rocks beyond the limitation of an unknown degree of W/R equilibration[21] (see Supplementary material). The variables controlling the isotopic compositions inherited from hydrothermally altered rocks are (a) hydrothermal fluid compositions; (b) initial mineral or rock composition; (c) temperature of interaction; and (d) fluid/rock ratios. When (b), (c), and (d) are constrained, the isotopic composition of hydrothermal fluid composition can be estimated by fitting mineral isotopic data with mixing trajectories calculated for variable degrees of water/rock interaction[21] (see Fig. 1).

The isotopic compositions of our samples are expressed in the conventional δ notation, where the $\delta^{17}O$ and $\delta^{18}O$ values are defined as follows:

$$\delta^{17}O = 10^3 \left[ (^{17}O/^{16}O)_{sample} / (^{17}O/^{16}O)_{standard} - 1 \right], \text{ and} \quad (1)$$

$$\delta^{18}O = 10^3 \left[ (^{18}O/^{16}O)_{sample} / (^{18}O/^{16}O)_{standard} - 1 \right], \text{ respectively} \quad (2)$$

A linearized form of the δ-notations is given by the δ'-notation[29], which is defined as follows:

$$\delta'^{17}O = 10^3 \ln \left[ (\delta^{17}O/1000) + 1 \right], \text{ and } \delta'^{18}O = 10^3 \ln \left[ (\delta^{18}O/1000) + 1 \right]. \quad (3)$$

The $\Delta'^{17}O$ value is defined as[30]:

$$\Delta'^{17}O = \delta'^{17}O - \lambda * \delta'^{18}O. \text{ The } \lambda \text{ value used in this work is } 0.528. \quad (4)$$

The $\Delta'^{17}O$ value of meteoric-hydrothermal water interacting with rocks covers a distinct range (Fig. 1) ranging from 0 to 0.06‰, averaging 0.03‰[30]. The isotopic equilibrium between quartz and water at hydrothermal temperatures (200, 300, and 400 °C, respectively) is calculated using the triple oxygen equilibrium fractionation factor between silica and water[29]. Unaltered igneous rocks have a narrow range of $\Delta'^{17}O$ values because the high-T θ value (where $\theta = \ln(\alpha^{17}O) / \ln(\alpha^{18}O)$ for minerals and melts) is close to the reference slope value of 0.528[22]. In contrast, hydrothermally altered rocks show significantly different values, due to the negative $\delta^{18}O$ value of the meteoric water responsible for the alteration and the extent of the water/rock ratio[21–24,31–33]. We chose initial $\Delta'^{17}O_{qz}$ values from $-0.052‰$ to $-0.090$, in the range of values for igneous rocks of continental and mantle source[22,23,31], in order to have the best fit through the measured data starting from the initial $\delta^{18}O_{qz}$ values of the unaltered facies of the lithotypes.

The targeted hydrothermal centres for triple oxygen isotope analysis in this study were the following: Oakley Glacier monzo-syenite (40 Ma); Mt McGee granite country rock and monzonite (38 Ma); Styx Glacier syenite (35 Ma); Cape King gabbro-diorite (34 Ma); Cape Crossfire monzo-diorite (34 Ma); No Ridge monzo-syenite (32 Ma)[25]. Additional details about samples are reported in Supplementary Information (SI_Samples).

Our dataset of $\delta^{17}O$ and $\delta^{18}O$ values (SI_Table 2) shows that the quartz of these intrusive rocks was altered from its original igneous value due to water/rock interaction. To estimate the oxygen isotope composition of primary meteoric waters, we interpolated the measured data of quartz from the hydrothermally altered rocks with mixing trajectories in the $\Delta'^{17}O$ - $\delta^{18}O$ space, for water-rock temperatures of 200, 300 and 400 °C. These temperatures encompass the range of temperature of hydrothermal water-rock interaction, interpreted to be -$300 \pm 50$ °C[34]. For each assumed $\delta^{18}O_{water}$ value, we calculated trajectories that fit through the measured $\delta^{18}O$ values of quartz. These trajectories intersect the triple oxygen isotope composition of rocks fully equilibrated with the assumed primary meteoric water. In the $\Delta'^{17}O$-$\delta^{18}O$ space, the data are fit by distinct curves (Fig. 1a–e), and the estimated $\delta^{18}O$ values for Eocene waters are in the range of: $-9.0 \pm 2‰$ at $40 \pm 0.6$ Ma (Oakley Gl.); $-14.5 \pm 2‰$ at $38 \pm 0.8$ Ma (Mt McGee); $-22.0 \pm 3‰$ at $35 \pm 1.75$ Ma (Styx Gl.); $-28.0 \pm 3‰$ at $35 \pm 2.5$ Ma (Cape Crossfire); $-30.0 \pm 2‰$ at $34 \pm 1.9$ Ma (Cape King); and $-19.0 \pm 1‰$ at $31.8 \pm 0.5$ Ma (No Ridge).

When plotted as calculated $\delta^{18}O_{water}$ values vs. time (Fig. 2), we see a large inflection point across the E-O boundary. Our data interpolation curve parallels the high-resolution curve of benthic forams[4] and shows the minimum $\delta^{18}O_{water}$ values at 34 Ma (provided the Ar$_{age}$ error limit). A similar although broad co-variation trend is observed for the compiled lowest δD values of meteoric waters (calculated from hydrothermal phases) of Tertiary intrusives[25]. This likely suggests that the isotopic signal depicted from meteoric hydrothermal waters closely reflects the variations of atmospheric conditions at the time of hydrothermal alteration. It is interesting to note that our data, similar to what is seen from the benthic foraminifera $\delta^{18}O$ curve, define a $\delta^{18}O_{water}$ increase back to higher values at -32 Ma. This period is reported to have experienced a rise in $pCO_2$ bringing the atmospheric temperatures back to warmer conditions[35].

Accurate evaluation of the compiled variations in MAT (mean atmospheric temperature) from south high latitude terrestrial proxies (south of 45°, most around 70°) defines atmospheric cooling to be in the range of 4 °C from 38 to 34 Ma, although offsets among temperature proxies up to 8 °C have been proposed[36]. Similarly, MAT proxy records from (mainly) northern mid-latitudes provide estimates of cooling from 0 °C to 8 °C[37]. As for global sea surface temperatures, there is increasing evidence of heterogenous temperature variations during late Eocene- early Oligocene based on oceanic proxy records,

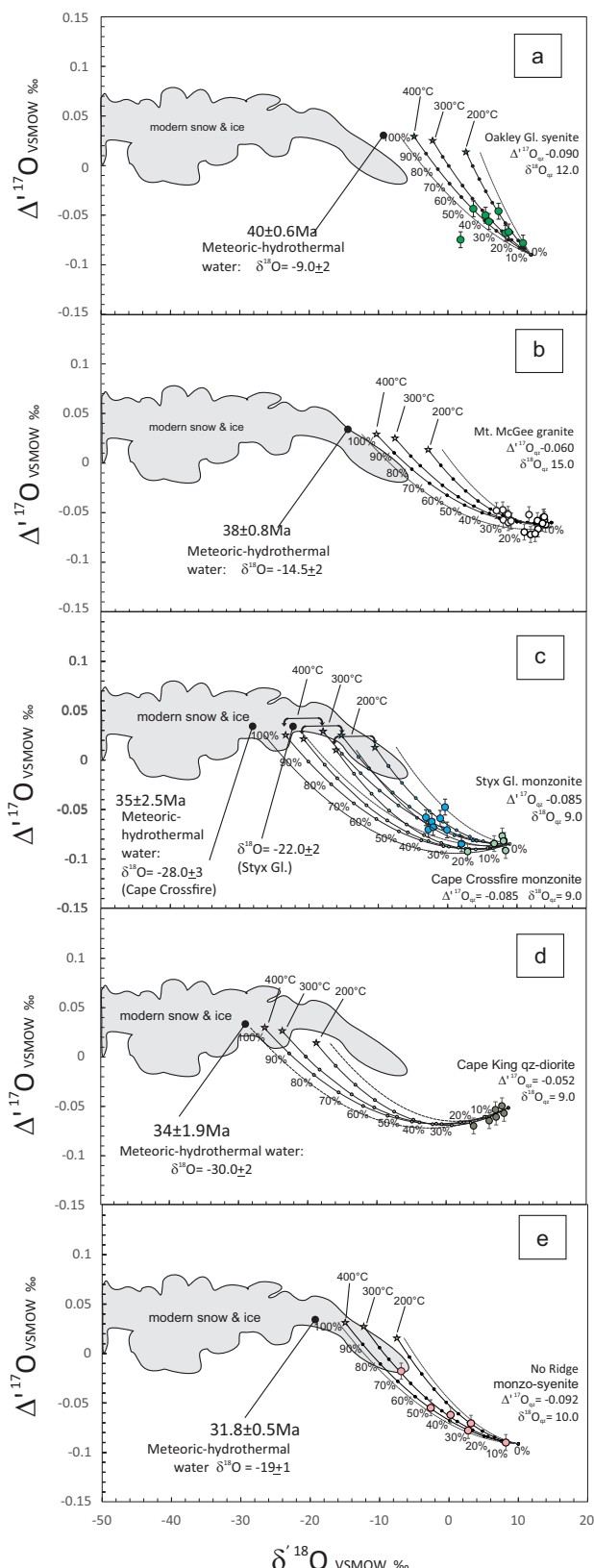

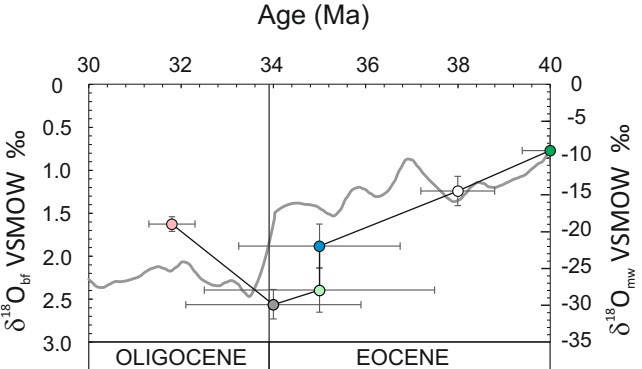

**Fig. 1 | a_e Mixing curves for triple oxygen isotope values.** Each panel (boxes **a**–**e**) represents a time interval and the related outcrop(s), namely: (**a**) Oakley Glacier syenite, 40 ± 0.6 Ma; (**b**) Mt. McGee granite, 38 ± 0.8 Ma; (**c**) Styx Glacier and Cape Crossfire monzonites, 35 ± 2 Ma; (**d**) Cape King qz-diorite, 34 ± 1.9 Ma; (**e**) No Ridge monzo-sienite, 31.8 ± 0.5 Ma. Mixing curves represent hydrothermal alteration of Antarctic igneous rocks for 200 °C, 300 °C, and 400 °C. The dots on the mixing curves define the percentages of W/R (water-rock) interaction. The stars indicate rock values for infinite W/R ratios. Starting $\delta^{18}O_{qz}$ values vary in the range of mantle and crustal rocks[22,23,31]: $\delta^{18}O_{qz}^{syenite}$ = 12 ‰, $\delta^{18}O_{qz}^{granite}$ = 15‰, $\delta^{18}O_{qz}^{monzonite}$ = 9‰, $\delta^{18}O_{qz}^{qz-diorite}$ = 9‰. The initial $\Delta^{17}O$ values are in the range of granitoid rocks[28]. The grey, dotted lines are also W/R mixing curves (same as the black lines), but they are the W/R curves that fit the varied model parameters reported in the model data (SI_Model Data). Their end also defines the infinite water value. The best-fit W/R mixing line defines the $\delta^{18}O_w$ of Eocene waters. The grey and the pale grey areas represent the composition of modern ice and snow, and surface and subsurface waters, respectively, according to [73]. Error bars report the maximum error derived from the lab internal standard, that is 0.3 ‰ for $\delta^{18}O$ and 8 ppm for $\Delta^{17}O$.

**Fig. 2 | $\delta^{18}O_{meteoric\ waters}$ vs. Age plot.** A clear variation at the Eocene-Oligocene boundary along a prolonged decreasing trend can be envisaged. The grey line is the long terms smoothed $\delta^{18}O_{benthic\ forams}$ vs. Age[4] (left $\delta^{18}O$ scale). The dark line is the interpolated line of the data reported in this study (right scale). Error bars of Ar-ages are from [25].

from marine proxies. Using modern local[39,40] and global[41] $\delta^{18}O/T(°C)$ lapse rates, the − 15.5‰ difference in calculated $\delta^{18}O_{waters}$ of hydrothermal systems at the 38 and 34 Ma, corresponds to a surface temperature decrease of ~ 19 ± 5 °C (based on modern $\delta^{18}O/T(°C)$ rate in Antarctica) and -12 ± 5.4 °C (based on modern $\delta^{18}O/T(°C)$ rate on the global scale), respectively. The values based on Antarctic $\delta^{18}O/T(°C)$ rates[39,40] exceed reconstructions based on oceanic proxies and likely indicate that modern high-latitude $\delta^{18}O/T(°C)$ slopes do not hold for absolute paleotemperatures. This may be due to to a combined effect of atmospheric temperature drop and changes in moisture circulation[42,43].

## Discussion

The $\delta D$ values for biotes and amphiboles of intrusive rocks from Cape King, No Ridge, Styx Glacier, and Cape Crossfire intrusions were measured and compiled along with literature data of rock samples from the same outcrops (SI_Table 4). We observe distinct compositional ranges among different hydrothermal systems, and variable H-isotope data within the same outcrop, the latter suggesting different degrees of water-rock interaction and isotopic equilibration. For $\delta D_{water}$ recalculation we discarded a few granite country rocks in the SE region of Mt McGee that are cut by a late mafic dikes ($\delta D$ = − 185 ± 5 ‰, age 34.7 ± 0.7 Ma[26]. Their $\delta D$ values (≤ − 170‰) are the lowest among those from the Mt McGee area, and suggest they may be related to a much later water-rock interaction event. Similarly, we discarded one sample (DF8, $\delta D$ = 216‰) from Oakley Gl., which displays evidence of oxidation[25].

To convert the $\delta D$ values of measured kaersutitic amphiboles and biotites into $\delta D$ values of meteoric-hydrothermal waters assuming an

with differences in the range of 10 ± 2 °C within the 38-34 Ma interval[4,38].

The stable isotope composition of meteoric waters along East Antarctica coastline constrains the hydrochemical cycle over the continental margin; thus, we can compare whether the atmospheric conditions inferred from coastal crystalline rocks are consistent with those

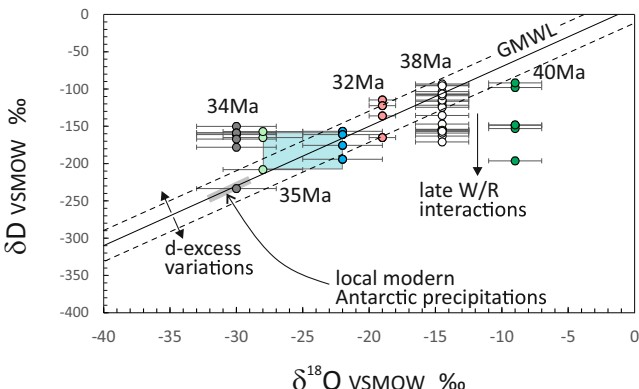

**Fig. 3 | Calculated stable isotope compositions of meteoric waters.** Meteoric waters that fed hydrothermal systems in Northern Victoria Land (NVL), Antarctica at the Eocene-Oligocene transition intercept the Modern Global Meteoric Water Line (GMWL). The range of Modern Antarctic precipitation is from[40]. Error bars on the X-axis represent the range of $\delta^{18}O_w$ values that fit the varied model parameters (SI_Model Data) and error bars on the Y-axis are the standard error of the mean for the measured $\delta D$ values.

equilibration temperature of 300 °C, we adopted 1000ln α $_{biotite-water}$ and 1000ln α $_{amphibole-water}$ values of + 20‰[44,45]. Due to the uncertainty in the hydrogen fractionation factor for Fe-rich amphibole, the calculated $\delta D$ values of meteoric hydrothermal waters may slightly change, but their relative H-isotope differences between the hydrous silicates and the hydrothermal water remain constant, allowing us to relative hydrogen isotope variations of meteoric waters through the Eocene. It has been recently questioned whether $\delta D$ values of hydrothermal waters may be accurately derived from the $\delta D$ values of hydrosilicates, because there may be continuous H-exchange between fluid and rock during cooling from peak hydrothermal temperatures[24,46]. Taking this point into consideration, we obtained O-isotope and H-isotope data independently; thus, we measured quartz for triple oxygen isotope ratios and we were able to calculate O-isotope composition of hydrothermal waters (SI_Analytical Methods). Then, we used the $\delta D$ values of the amphibole/biotite to derive the corresponding hydrogen isotope composition of the water. We used the calculated $\delta D_{water}$ and $\delta^{18}O_{water}$ values to estimate the stable isotope compositions of Cenozoic meteoric waters. Our data show that the oldest samples have a larger range of $\delta D_{water}$ values (− 92 > $\delta D$ > −196‰ at 40 ± 0.6 Ma; − 69 > $\delta D$ > −171‰ at 38 ± 0.8 Ma) than younger samples (− 154 > $\delta D$ > − 194‰, at 35 ± 2.5 Ma; − 150 > $\delta D$ >−234‰, at 34 ± 1.9 Ma; − 115 > $\delta D$ >−165‰, at 31.8 ± 0.5 Ma), and significantly higher $\delta^{18}O$ values (− 9 ± 2 and − 14.5 ± 2‰ at 40 and 38 Ma, respectively) than younger samples (− 22 ± 3; − 30 ± 2; − 19 ± 1‰, at 35, 34, and 32 Ma, respectively). Similar $\delta^{18}O$ and $\delta D$ values have been reported for average precipitations ($\delta D_{water}$ values between − 113 and − 151‰) of the Antarctic Peninsula at 35.9 ± 1.1 Ma[47].

Coupled $\delta D$ - $\delta^{18}O$ values define different compositional ranges for the samples at 40-38 Ma, and 34-32 Ma, representing how meteoric waters varied through the Eocene-Oligocene transition (Fig. 3). They intersect the modern GMWL, although significantly higher $\delta^{18}O$ values are shown by Cenozoic meteoric waters compared to the modern (i.e., modern snow samples at Styx Gl. are up to 10‰ lighter[40]. This implies that the modern GMWL possibly holds for the geological past, at least since the middle Cenozoic, which was a warm period thought to be particularly humid[48,49]. Accordingly, relative humidity, air masses temperature, and temperature of the oceanic source area of the precipitation controlled the deuterium-excess parameter (d = $\delta D$ − 8*$\delta^{18}O$)[50] for this period. Modern snow precipitations in East Antarctica have a d-excess variability from − 20 to + 29‰[51]; thus a variability up to a ± 30‰ range in d-excess may be expected also for the geological past, as a consequence of variable moisture source and water vapour transport during changing atmospheric conditions. In fact, the d-excess in marine water vapour inversely correlates with moisture source relative humidity, due to the kinetic fractionation during evaporation from the ocean into the atmosphere[52], and positively correlates with sea-surface temperature, due to the isotopic fractionation between seawater and water vapour[50,53,54]. During the late Eocene, moisture decrease has been inferred for the Antarctic Peninsula[55] and Greenland[56] based on paleo-botanical data and terrestrial biomarkers preserved in marine sediments. This is in agreement with the variations in weathering regime along the continental coastline of Antarctica, from mainly chemical, typical of warm and humid climate, to physical, characteristic of arid and cold temperature conditions, recorded in the Ross Sea and the Weddel Sea areas[57,58]. Our data do not depict any large d-excess variations, because short-term differences would be averaged at the My-time scale resolution of our research. However, it is reasonable to hypothesize that variations of d-excess in the order of ± 30‰ (or more) may have occurred in a geological time interval likely characterised by large differences in atmospheric moisture. This would also help to explain the low $\delta D$ values slightly above or below the modern GMWL (Fig. 3).

Theoretical estimates of moisture poleward transport suggest that the $\delta^{18}O$ and $\delta D$ values of meteoric precipitations are more influenced by atmospheric water sources rather than local conditions[59], so Antarctic coastal data should reflect the combined effects of atmospheric conditions and oceanic circulations. The difference in $\delta^{18}O_{water}$ values based on our data with those based on $\delta^{18}O$ values of benthic foraminifera may indicate that Eocene/Oligocene Antarctic moisture sources were situated at lower latitudes, allowing for a larger distillation along the moisture pathways, and eventually changing the transport characteristics (i.e., eddy fluxes vs. advection)[59]. The onset of a "proto" Antarctic Circumpolar Current (ACC) also caused significant changes in the circulation patterns at high latitudes, implying that moisture origin, transport and supply to northern Victoria Land coastal areas might have been strongly modified[60–67]. Due to the ice cap extension beyond the polar circle and variations of oceanic currents, the vapour source of Antarctic precipitations may have been "pushed" towards low latitudes.

Transport modelling of modern precipitation predicts that moisture source latitude is a function of altitude and distance from the coastline. East Antarctica moisture sources are presently at latitudes between 46° and 50°_S. Interestingly, Victoria Land presently has a moisture sources at even lower latitudes (i.e., 42°_S)[68]. Because the water-vapour fractionation factor increases with decreasing temperatures, the residual water vapour resulting from the Rayleigh condensation processes becomes increasingly depleted in oxygen and hydrogen isotope composition with distance from the source. Moreover, the opening of the Southern Ocean gateways (i.e., the Drake Passage and the Tasmanian Gateway) produced a cooling of the Southern Ocean and a reduction of poleward heat transport[60,61]. Thus, a progressively increasing temperature gradient was established from the equator to the poles[69], likely resulting in less evaporation at higher latitudes. It is therefore possible that Antarctic meteoric waters before the E-O transition had been more influenced by precipitation generated in proximity of the coastline.

If so, the rearrangement of oceanic currents and the shift of atmospheric temperatures along Antarctic margins induced irreversible changes in the hydrological regime of the coastal areas during the Eocene-Oligocene climatic transition. Temperature-induced isotopic variation possibly occurred in conjunction with a reduction in moisture availability and/or a variation of moisture source areas, which resulted in low $\delta^{18}O$ values of meteoric waters after the E/O transition. This would also explain why the $\delta^{18}O$ values that we reconstruct are even lower than what we would expect from the paleo-temperature reconstructions.

## Methods

Mineral separates of quartz were obtained by crushing and sieving the rock sample, followed by standard magnetic separation and hand-picking under the binocular microscope. Oxygen isotope compositions were measured using laser fluorination[70], and hydrogen isotope compositions were determined using high-T reduction[71]. All isotope data are reported relative to the VSMOW-SLAP2 scale using internal standards that were directly calibrated to VSMOW2 and SLAP2[72].

## Data availability

The data generated in this study (Supplementary Table 1,2,4,5, and Model Data) are provided in the Supplementary Information.

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

## Acknowledgements

L.D. is grateful to the PNRA for logistics support in Antarctica and to Ray Burgess for sharing fieldwork. L.D. thanks Museo Nazionale dell'Antartide for providing rocks samples. L.D. is also grateful to Viorel Atudorei, Erick Cano, and Tommaso Di Rocco for their help in analytical work. The research was supported by CNR-Short Term Mobility Programme n. 0063613/2018.

## Author contributions

L.D. conceived the research, sampled the rocks and performed the analyses; L.D. and Z.D.S. discussed the data, made the modelling, and wrote the manuscript.

## Competing interests
The authors declare no competing interests.
