## [Peer Review File · Nature Communications]

A tipping point in stable isotope composition of Antarctic meteoric waters during Cenozoic glaciationREVIEWER COMMENTS

Reviewer #1 (Remarks to the Author):

Review for Dallai and Sharp (NCOMMS-23-49350) entitled: A tipping point in stable isotope composition of meteoric waters during the onset of the Cenozoic glaciation in Antarctica.

General comment:

The water isotopic composition of polar ice cores is the most informative climate record today, but ice much older than 1 Ma is not preserved. Reconstructing the isotopic composition of paleo-water from other material is challenging, but the most direct way to infer information on the paleo-hydrosphere. The authors derive the water isotopic composition of 34 to 40 Ma old meteoric waters from hydrothermally altered rocks collected in Antarctica. The specific application and the combination with hydrogen isotopes is novel, so this paper provides a benchmark for future studies. I support publication of the manuscript after some modifications as suggested below.

Major comments:

The errors derived from the analyses of the lab internal standard is 0.3 permill for d18O and 8 ppm for D17O. These errors must be displayed in Figure 1 as typical errors and stated in the figure caption or in the main text.

The authors simply say that all data are reported in SMOW-SLAP scale, but only three analyses of an international silicate standard NBS28 are reported. Please expand on this. Report e.g. the silicate standard values used to for calculating the SMOW-SLAP scale. The silicate reference frame should be clear.

The Supplementary Table 3 reports 'Model curve data', but errors are missing. At least report the exact formulae used to derive these values. They are your core results, and it must be clear how they are derived. In this context, Figure 1 should be expanded and also show the meteoric water line as well as the theoretical 100% alteration endmember for qz that is used to determine the unaltered paleo-water isotopic compositions.

What criteria are used to define the 'best fit'? I'm a bit puzzled with Fig. 1 panels e and f. The w/r ratios are very low, which should result in large errors in the extrapolated d18O of meteoric water. Yet in panel 'f' (i.e. cape king) the smallest error of only 2 permill is reported. This appears optimistic. In panel 'e' the entire curve seems to be dominated by one sample (D2), which has a different sampling name than all the other samples (all ARxx) and it seems that D2 was only analyzed once (no error; or is '1s' the error of a single gas measurement?). In general, please provide the number of replicates analyzed (n) in the table. If this was my paper, I would invest some more measurements to make sure that the D2 datapoint is analytically robust. These two sample sites ('e' and 'f') give the lowest d18O of meteoric water (both at 34 Ma) and seem to confirm each other. However, considering the low w/r ratios, I am most skeptical of those two. I cannot help noticing that the story would simplify if those two values were actually a bit higher. This is not only true for the general trend, which would be more consistent with the foram record (see Figure 2), but also for the correlation between d18O and dD, which are two independent estimates. They would become more consistent if the d18O for the 34Ma samples was actually higher (i.e. yellow and dark blue datapoints moved to the right in Fig. 3 – please plot the error bars in this figure). It does not seem simple to explain dD values above the GMWL in Fig. 3 and a somewhat biased d18O record (or larger errors due to low w/r ?) for the two samples with the lowest w/r is probably one viable solution here.

Despite all of this very open and direct (german style) criticism, I strongly support publishing this paper. The authors demonstrate the feasibility of an innovative approach that is especially suitable for polar regions, where the isotopic difference between the silicate rock and the interacting water is large. When excluding the two 34Ma sample sites, the overlap with the traditional foram record is fantastic. Part of the way forward is to find the samples with the highest w/r ratios and more sites, which is possible with more field work. Reading the paper sparked some new ideas how to identify potential problems such as later hydrothermal overprint. So while I maintain some healthy skepticism concerning the present results, I see a lot of potential for this approach.

Minor comments:

Title: composition of

First paragraph: The sentence starting "Meteoric-hydrothermal water ..." makes no sense to me. Do you mean imprinted their stable isotopic composition on plutonic rocks?

Line six: "the the"

It is unclear if the initial isotopic composition of the plutonic rocks is directly measured or only assumed. Please clarify.

Fig. 1: It should be $D'17O$ on the y-axis, not $D'18O$.

Fig. 1. I was not sure what the orange and what the brown line is. The orange could be light brown and the brown could be dark orange. Use something more distinct.

Fig. 1. Write percentages on at least one line.

Fig. 1: The y-axis stops at 0. Include the meteoric water line and the 100% alteration endmember of qz, as well as the water estimates from that qz endmember for the respective temperature range. As it is, the Figure is not clear to the average reader.

Figure Caption 1: The authors write: "The values of the meteoric waters are independent from W/R ratios because these affect the triple oxygen isotope composition of the samples only." This is unclear and clearly contradicts what is written in the introduction on "Interaction of meteoric waters with intrusive rocks". I would say the sentence is wrong.

I think the authors mix up Barkan and Luz 2007 and Luz and Barkan 2010. The two papers are not cited for the appropriate sentences on page 4.

Page 4: Either use $D'17O$ or 17O-excess. Not both.

You cite important papers. Maybe include Zakharov et al. 2019 in chem geol. This is the only one that combines triple oxygen isotopes with hydrogen isotopes and in contrast to the others studies a modern hydrothermal system (Iceland).

I hope the authors find my comments constructive,
Daniel Herwartz

Reviewer #2 (Remarks to the Author):

Review of "A tipping point in stable isotope composition of meteoric waters during the onset of Cenozoic glaciation in Antarctica" by Dallai and Sharp, submitted to Nature Communications.

Dallai and Sharp reconstruct the isotopic composition of meteoric waters using hydrothermally altered, Antarctic plutonic rocks of the Eocene and Oligocene. They use triple oxygen isotope geochemistry and water-rock mixing models to reconstruct the $d18O$ of meteoric water. They separately measure dD in different mineral phases and combine this dD with $d18O$ to compare paleo-meteoric waters to the modern global meteoric water line. They report a decline in $d18O$ of meteoric waters from 42 to 34 Ma, and then an increase from 34 to 32 Ma. Many of their Eocene-Oligocene $d18O$ - dD reconstructions fall on or near the GMWL.

The presentation of this work is very sloppy in its current state and unacceptable for publication. The figures need significant work; in particular, Figure 1 is illegible. The text is poorly organized and contains numerous typographic, grammatical, and formatting errors. I started commenting on them but quickly gave up. Please edit with an eye for details. The supplement is missing information on the water-rock mixing model.

I want this paper to be good because triple oxygen isotope geochemistry is an emerging field, and everyone benefits from well-presented, thoughtfully discussed datasets so that future similar datasets can be accepted. This is a unique dataset and potentially a very compelling story given the interesting inflection point in $d18O$ meteoric water after the EOT. It would be ideal to see one more data point at

30 Ma, rock record willing. To make it publishable, it needs: 1) a more direct link between the data and the discussion -- possibly an additional figure illustrating the hydrologic mechanisms describe, 2) potential re-write/re-organization, as is the paper bounces from results to discussion and back, 3) figures need to improved (some suggestions below).

Introduction: you could improve this by stating a specific need/question that your samples can address. You end the introduction with the idea that " Eocene meteoric waters should have varied in response of differences in the global hydrological cycle." I think you could be more specific in terms of directionality and what differences you are trying to understand. Furthermore, you spend significant time/text in discussion temperature differences, but there is almost no motivation in the introduction framing those ideas.

Missing line numbers! Line-by line comments below given based on page # and position.

Make sure that you use a prime rather than an apostrophe throughout the text and figures when referring to $\delta^{18}O$, $\Delta^{17}O$, etc.

page 2 response to differences

page 3 near the top: rocks of what lithology? It's important to mention this early on

page 3 near the bottom: why is the squiggly equals signed being used instead of =?

add citation for assuming $\lambda = 0.528$

page4: "We used -0.052 ‰ for the starting value of gabbro-diorite quartz, similar to mafic rocks, and $-0.085 \pm 0.01 \text{ ‰}$ for more acidic intrusives, similar to granites, respectively."

Clarify that you mean D17O values. No error on the value for gabbro-diorite quartz? What is the reference for these values??

page 4 near the top: you switch to $17O_{excess}$ notation. Be consistent throughout.

page 4 near the top: change in in-text referencing format

page 5 near the top: " Our data interpolation curve parallels the high resolution curved of benthic forams (Westerhold et al. 2020) and shows the minimum temperatures at 34Ma provided the Arage error limit ($\sim 5\%$) " temperature is not plotted in Fig 2. Revise. You are linking T with d18O implicitly before explaining that relationship.

page 5 You use the phrase 'temperature variations' when I think you mean temperature difference through your record.

bottom of page 5- What are the δ/T lapse rates used in these calculates? What is the temperature change based on the reconstruction from oceanic proxies? restate here.

Page 6 - This is the first mention of which minerals you are measuring dD in. This information should be stated earlier.

Page 7 - Plot d-excess in Figure 3 as a second panel. This will help the reader follow along with your discussion. I disagree with this sentence " Our data do not depict any large d-excess variations." I don't know what you mean by "large," but to my eye, many of your samples plot significantly above and below the GMWL.

Evaluation  evolution (?)

Supplementary tables 1 and 2: you need to report d18O and d17O to the third decimal place to enable calculations of D17O.

Supplementary table 3: The table formatting nee. What does non-prime refer to?

Page 6 bottom - While I agree that the Cenozoic GMWL is likely similar to that of modern, your data do not necessarily support that conclusion. Becuase of the d18O-dD spread in your samples at each location, one could draw MWLs with slopes very different from 8 and still "intersect" all of your samples.

Page 7-8: I find this discussion interesting but as written it does not clearly relate to the data presented. Please make explicit connections between the mechanisms you are invoking and how they explain and/or are supported by the data.

End of Page 8 " Temperature-induced isotopic variation possibly occurred in conjunction with a reduction in moisture availability and/or a variation of moisture source areas, which resulted in low $\delta^{18}O$ values of meteoric waters. " low d18O values in which time period? I think you mean during 34

Ma. How do you explain the increase back to higher $\delta^{18}\text{O}$ values at 32 Ma?

Figure 1: The blue overlays make this figure impossible to read. Were the overlays supposed to be transparent? I cannot see the data points, tick marks, or the mixing lines described in the legend. The y axis is labeled incorrectly. Inconsistent use of () in the axes labels compared to Figure 2 and 3.

Figure 2: The x-axis continues past data points. Either adjust the x-axis so it stops at 30Ma or continue the benthic foram record. It might be useful to indicate the geologic periods/the EOT on here explicitly.

Figure 3: The axes labels are floating in space.

The caption has grammatical and typographic errors.

How are the late W-R samples identified? It looks like an arbitrary rectangle in this figure, and this is not explained in the text.

Reviewer #3 (Remarks to the Author):

This manuscript presents $\delta^{18}\text{O}$ and δD values of meteoric waters for the late Eocene to early Oligocene of Antarctica derived from the geochemistry of crystalline rocks. The manuscript presents interesting data and ideas and is broadly well written, albeit with some instances where minor clarification is required. Some minor clarifications are also required for the figures. I am not an expert in crystalline rock geochemistry so am unable to comment in an informed manner on the systematics or methods of this, so have focused on the interpretation and paleo-environmental discussion. The key result of a marked shift in the hydrology of the continent at the EOT is an interesting and important finding, but it is important to see these results better integrated and compared to previous studies which have looked at this using plant wax δD and other paleo-environmental information. See more specific comments below.

I recommend including line numbers in future submissions for ease of review.

Introduction: towards the end of the first paragraph, 'affecting continental and oceanic water balance'. Can you clarify what you are meaning by this phrase.

'SST were still high at the Eocene-Oligocene transition'. The references included here primarily refer to CO_2 trends through this interval, but there is evidence for cool Antarctic-proximal temperatures by the late Eocene i.e. see papers mentioned below.

I suggest adding a comment into the sample description on how these samples were dated. Even though this is in the supplementary, I'd recommend a comment up front especially as you refer to the error on these age estimates later in the text.

Last paragraph of 'Interaction of meteoric waters with intrusive rocks'. Second sentence- when you refer to 'minimum temperatures at 34 Ma' from your data, what temperature is this referring to? Air? You explain later in the paragraph how you calculate temperature from your data, so I'd recommend some rearranging here to improve clarity.

Second sentence in that paragraph. I recommend a more recent reference here, as a lot of proximal proxy temperature records have been published around the continent in the last decade i.e. as summarised in Tibbett et al. 2023, *Paleoceanography and Paleoclimatology*, and a recent Ross Sea record close to the sites in this study in Duncan et al., 2022, *Nature Geoscience*. Tibbett et al. (2023) is a useful comparison for your numbers here, as their compiled SST proxies cool by up to 3°C and MAT by up to 4°C between the late Eocene and early Oligocene, and so support your statement that using modern $\delta/T^\circ\text{C}$ rates reconstructs a temperature difference greater than seen in the proxy records. However, I would alter the wording in your abstract where you state that your results find higher temperature variations than recorded by ocean proxies, as you discuss later that this is likely due to the modern relationship being unsuitable to use as an inference for absolute temperature change. A 19°C difference is certainly much larger than would be supported by a wide range of both proxy SST and MATs as well as other paleo-environmental information.

There are several key references missing here that would inform and largely support your findings based on δD records preserved in plant waxes (a Miocene discussion in Feakins et al., 2012 *Nature Geoscience*, and EOT discussions in Feakins et al., 2014, *EPSL* and Tibbett et al., 2021, *Paleoceanography and Paleoclimatology*). In particular Feakins et al., 2014 use isotope enabled

modelling to look at precipitation over the EOT, and forms an important comparison for this work.
I recommend having a location figure in the main text.

Figure 1b: what do the two different colour sample points represent?

Figure 2: If the blue record is the benthic foraminifera record from Westerhold et al., 2020 then this should have a secondary axis to show the actual values.

Figure 3: Add the definition of WR to figure caption.

Reference 23 needs a title

Supplementary:

Fig1.S1: Please add a box on the inset to show what part of Antarctica the main map represents. Does the grey just represent exposed rock rather than ice? Or a specific type of bedrock?

Point to Point revision for Dallai and Sharp (NCOMMS-23-49350) entitled: A tipping point in stable isotope composition of meteoric waters during the onset of the Cenozoic glaciation in Antarctica.

Reviewer #1:

General comment:

The water isotopic composition of polar ice cores is the most informative climate record today, but ice much older than 1 Ma is not preserved. Reconstructing the isotopic composition of paleo-water from other material is challenging, but the most direct way to infer information on the paleo-hydrosphere. The authors derive the water isotopic composition of 34 to 40 Ma old meteoric waters from hydrothermally altered rocks collected in Antarctica. The specific application and the combination with hydrogen isotopes is novel, so this paper provides a benchmark for future studies. I support publication of the manuscript after some modifications as suggested below.

Major comments:

The errors derived from the analyses of the lab internal standard is 0.3 permill for d18O and 8 ppm for D17O. These errors must be displayed in Figure 1 as typical errors and stated in the figure caption or in the main text. The error bars were inserted in Figure 1. The error on the X-axes is within the symbols.

The Supplementary Table 3 reports 'Model curve data', but errors are missing. At least report the exact formulae used to derive these values. They are your core results, and it must be clear how they are derived. The complete formulas are now reported in the Supplementary files.

In this context, Figure 1 should be expanded and also show the meteoric water line as well as the theoretical 100% alteration endmember for qz that is used to determine the unaltered paleo-water isotopic compositions. Figure 1 has been redrawn: the field of modern snow, ice, and precipitations (Aaron et al., 2021) has been added. Also the percentage of w/r interaction along the exchange curves has been inserted and the 100% alteration endmember highlighted (star symbol).

What criteria are used to define the 'best fit'? I'm a bit puzzled with Fig. 1 panels e and f. The w/r ratios are very low, which should result in large errors in the extrapolated d18O of meteoric water. Yet in panel 'f' (i.e. cape king) the smallest error of only 2 permill is reported. This appears optimistic. In panel 'e' the entire curve seems to be dominated by one sample (D2), which has a different sampling name than all the other samples (all ARxx) and it seems that D2 was only analyzed once (no error; or is '1s' the error of a single gas measurement?). The curves are calculated in order to fit through all data points within their error bars.

In general, please provide the number of replicates analyzed (n) in the table. If this was my paper, I would invest some more measurements to make sure that the D2 datapoint is analytically robust. These two sample sites ('e' and 'f') give the lowest d18O of meteoric water (both at 34 Ma) and seem to confirm each other. However, considering the low w/r ratios, I am most skeptical of those two. I cannot help noticing that the story would simplify if those two values were actually a bit higher. This is not only true for the general trend, which would be more consistent with the foram

record (see Figure 2), but also for the correlation between $\delta^{18}\text{O}$ and δD , which are two independent estimates. They would become more consistent if the $\delta^{18}\text{O}$ for the 34Ma samples was actually higher (i.e. yellow and dark blue datapoints moved to the right). We agree with the reviewer's comments, but we simply reported the measured data: so, the curves were calculated to fit the datapoints. The scatter of data may be interpreted also in view of the geochronological error. The number of replicates was inserted in the supplementary data files. As for the sample D2, the label corresponds to the initials of the sampler (Dallai in this case). The AR samples were collected in a different campaign, but in the same outcrop.

in Fig. 3 – please plot the error bars in this figure). It does not seem simple to explain δD values above the GMWL in Fig. 3 and a somewhat biased $\delta^{18}\text{O}$ record (or larger errors due to low w/r ?) for the two samples with the lowest w/r is probably one viable solution here. The error bars were inserted in Figure 3. The bar on the X-axis defines the range of values inferred for the $\delta^{18}\text{O}_{\text{H}_2\text{O}}$ at each locality/age. The bar on the Y-axis is $\pm 5\%$, that is the maximum standard error of the mean for each measured sample.

Despite all of this very open and direct (german style) criticism, I strongly support publishing this paper. The authors demonstrate the feasibility of an innovative approach that is especially suitable for polar regions, where the isotopic difference between the silicate rock and the interacting water is large. When excluding the two 34Ma sample sites, the overlap with the traditional foram record is fantastic. Part of the way forward is to find the samples with the highest w/r ratios and more sites, which is possible with more field work. Reading the paper sparked some new ideas how to identify potential problems such as later hydrothermal overprint. So while I maintain some healthy skepticism concerning the present results, I see a lot of potential for this approach.

Minor comments:

Title: compositionof - corrected

First paragraph: The sentence starting “Meteoric-hydrothermal water ...” makes no sense to me. Do you mean imprinted their stable isotopic composition on plutonic rocks? - corrected accordingly

Line six: “the the” – [line 24]: corrected

It is unclear if the initial isotopic composition of the plutonic rocks is directly measured or only assumed. Please clarify – [lines 107-109]: The used $\Delta^{17}\text{O}_{\text{qz}}$ values vary from -0.050% to -0.090% , that is in the range of values for igneous rocks of continental and mantle source (Pack and Hewartz, 2014; Sharp et al., 2018; Zakharov and Bindeman, 2019). They were chosen in order the curve to have the best fit through the measured data.

Fig. 1: It should be $\text{D}'^{17}\text{O}$ on the y-axis, not $\text{D}'^{18}\text{O}$. - corrected

Fig. 1. I was not sure what the orange and what the brown line is. The orange could be light brown and the brown could be dark orange. Use something more distinct. - Colors are now uniform for the trajectories of a same outcrop, and the temperatures of w/r exchange are reported for each curve.

Fig. 1. Write percentages on at least one line. - corrected

Fig. 1: The y-axis stops at 0. Include the meteoric water line and the 100% alteration endmember of qz, as well as the water estimates from that qz endmember for the respective temperature range. As it is, the Figure is not clear to the average reader. - modified

Figure Caption 1: The authors write: "The values of the meteoric waters are independent from W/R ratios because these affect the triple oxygen isotope composition of the samples only." This is unclear and clearly contradicts what is written in the introduction on "Interaction of meteoric waters with intrusive rocks". I would say the sentence is wrong. – the sentence has been removed, as we it was actually wrong, as correctly pointed out by the reviewer.

I think the authors mix up Barkan and Luz 2007 and Luz and Barkan 2010. The two papers are not cited for the appropriate sentences on page 4. – [line 100]: the correct ref (Luz and Barkan, 2010) is now reported.

Page 4: Either use $\Delta^{17}\text{O}$ or 17O -excess. Not both. – [line 99]: the notation was homogenized to $\Delta^{17}\text{O}$

You cite important papers. Maybe include Zakharov et al. 2019 in chem geol. This is the only one that combines triple oxygen isotopes with hydrogen isotopes and in contrast to the others studies a modern hydrothermal system (Iceland). – inserted at [line 106] and Refs.

I hope the authors find my comments constructive,
Daniel Herwartz

Reviewer #2 (Remarks to the Author):

Review of "A tipping point in stable isotope composition of meteoric waters during the onset of Cenozoic glaciation in Antarctica" by Dallai and Sharp, submitted to Nature Communications.

Dallai and Sharp reconstruct the isotopic composition of meteoric waters using hydrothermally altered, Antarctic plutonic rocks of the Eocene and Oligocene. They use triple oxygen isotope geochemistry and water-rock mixing models to reconstruct the $d_{18}\text{O}$ of meteoric water. They separately measure $d\text{D}$ in different mineral phases and combine this $d\text{D}$ with $d_{18}\text{O}$ to compare paleo-meteoric waters to the modern global meteoric water line. They report a decline in $d_{18}\text{O}$ of meteoric waters from 42 to 34 Ma, and then an increase from 34 to 32 Ma. Many of their Eocene-Oligocene $d_{18}\text{O}$ - $d\text{D}$ reconstructions fall on or near the GMWL.

The presentation of this work is very sloppy in its current state and unacceptable for publication. The figures need significant work; in particular, Figure 1 is illegible. - All figures were redrawn taking into consideration reviewers' comments.

The text is poorly organized and contains numerous typographic, grammatical, and formatting errors. I started commenting on them but quickly gave up. Please edit with an eye for details. –Text was reorganized and checked for typographic, grammatical, and formatting errors.

The supplement is missing information on the water-rock mixing model. -The mixing model and equations are now reported in detail in Supplementary Information

I want this paper to be good because triple oxygen isotope geochemistry is an emerging field, and everyone benefits from well-presented, thoughtfully discussed datasets so that future similar datasets can be accepted. This is a unique dataset and potentially a very compelling story given the interesting inflection point in $\delta^{18}\text{O}$ meteoric water after the EOT. It would be ideal to see one more data point at 30 Ma, rock record willing. - Preliminary geo-chronological data indicate that the alkaline intrusives along the Ross Sea coastline intruded till 25 Ma. This provide the possibility to further investigate. However, available samples do not allow a triple-oxygen isotope study.

To make it publishable, it needs: 1) a more direct link between the data and the discussion -- possibly an additional figure illustrating the hydrologic mechanisms describe,

2) potential re-write/re-organization, as is the paper bounces from results to discussion and back, 3) figures need to improved (some suggestions below).

Introduction: you could improve this by stating a specific need/question that your samples can address. You end the introduction with the idea that " Eocene meteoric waters should have varied in response of differences in the global hydrological cycle." I think you could be more specific in terms of directionality and what differences you are trying to understand. Furthermore, you spend significant time/text in discussion temperature differences, but there is almost no motivation in the introduction framing those ideas.

Missing line numbers! Line-by-line comments below given based on page # and position. - line numbers inserted.

Make sure that you use a prime rather than an apostrophe throughout the text and figures when referring to $\delta^{18}\text{O}$, $\Delta^{17}\text{O}$, etc. – the text was checked to ensure the prime-notation was used.

page 2 response to differences – [line 57]: corrected

page 3 near the top: rocks of what lithology? It's important to mention this early on – [lines 65]: it was specified “ alkaline intrusive rocks (diorite to syenite)”

page 3 near the bottom: why is the squiggly equals signed being used instead of = [line 93-94]: the \approx sign was changed to = according to the usual notation, although it is the approximation formula of Taylor-McLaurin.

add citation for assuming $\lambda = 0.528$ – the reference was inserted [ine 98]

page4: "We used -0.052 ‰ for the starting value of diorite quartz, similar to mafic rocks, and -0.085 \pm 0.01 ‰ for more acidic intrusives, similar to granites, respectively."

Clarify that you mean $\Delta^{17}\text{O}$ values. No error on the value for gabbro-diorite quartz? What is the reference for these values?? [lines 107-109]: The used $\Delta^{17}\text{O}_{\text{qz}}$ values vary from -0.050 ‰ to -0.090, that is in the range of values for igneous rocks of continental and mantle source (Pack and Hewartz, 2014; Sharp et al., 2018; Zakharov and Bindeman, 2019). They were chosen in order the curve to have the best fit through the measured data.

page 4 near the top: you switch to ^{17}O excess notation. Be consistent throughout. –[line 103]: corrected.

page 4 near the top: change in in-text referencing format [line 102]: the wrong format reference was corrected

page 5 near the top: " Our data interpolation curve parallels the high resolution curved of benthic forams (Westerhold et al. 2020) and shows the minimum temperatures at 34Ma provided the Arage error limit (~ 5%) " temperature is not plotted in Fig 2. Revise. You are linking T with d18O implicitly before explaining that relationship. [lines 130]: the sentence was re-phrased in order to correctly express the comparison between our $\delta^{18}\text{O}_{\text{water}}$ data and the $\delta^{18}\text{O}_{\text{benthic forams}}$

page 5 You use the phrase 'temperature variations' when I think you mean temperature difference through your record. –[line 138]: the word “variations” was changed to “differences”

bottom of page 5- What are the δ/T lapse rates used in these calculates? What is the temperature change based on the reconstruction from oceanic proxies? restate here. – [lines 149 to 152]: The δ/T rates are quoted in the text, and the data are reported in the supplementary file.

Page 6 - This is the first mention of which minerals you are measuring dD in. This information should be stated earlier. – [line 161]: biotites and amphiboles minerals (measured for their H-isotope compositions) were specified at the very beginning of the paragraph.

Page 7 - Plot d-excess in Figure 3 as a second panel. This will help the reader follow along with your discussion. I disagree with this sentence " Our data do not depict any large d-excess variations." I don't know what you mean by "large," but to my eye, many of your samples plot significantly above and below the GMWL. - The sentence was changed in the text [line: 201] and the possibility of a d-excess variation in the + 30 permil range (as observed in the modern times) was introduced.

Evaluation  evolution (?) - [line 217]: the word was changed with “estimates”

Supplementary tables 1 and 2: you need to report d18O and d17O to the third decimal place to enable calculations of D17O. - corrected

Supplementary table 3: The table formatting nee. What does non-prime refer to? – the table was re-formatted and the δ -values reported are those used for the calculations and the discussion.

Page 6 bottom - While I agree that the Cenozoic GMWL is likely similar to that of modern, your data do not necessarily support that conclusion. Becuase of the d18O-dD spread in your samples at each location, one could draw MWLs with slopes very different from 8 and still "intersect" all of your samples. The data spread may due to incomplete H-isotope reset at low w/r ratios. However, the possibility exist that the d-excess parameter may have varied in the past according to the arid vs. humifid climate states during the greenhouse-to-icehouse transition. The modern GMWL is able to intesect most data, suggesting that Cenozoic GMWL may be likely similar to the modern one.

Page 7-8: I find this discussion interesting but as written it does not clearly relate to the data presented. Please make explicit connections between the mechanisms you are invoking and how they explain and/or are supported by the data. The text was modified in its rationale in order to

relate more adequately to the data presented.

End of Page 8 " Temperature-induced isotopic variation possibly occurred in conjunction with a reduction in moisture availability and/or a variation of moisture source areas, which resulted in low $\delta^{18}\text{O}$ values of meteoric waters. " low $\delta^{18}\text{O}$ values in which time period? I think you mean during 34 Ma. How do you explain the increase back to higher $\delta^{18}\text{O}$ values at 32 Ma? – [lines 134-135]: It is interesting to note that our data, similar to what displayed from the bethic foraminifa $\delta^{18}\text{O}$ curve, depict a $\delta^{18}\text{O}_{\text{water}}$ increases back to higher values at 32 Ma. This period is considered to experience a new rise in $p\text{CO}_2$ bringing the atmospheric temperatures back to warmer conditions (Pagani et al, 2005).

Figure 1: The blue overlays make this figure impossible to read. Were the overlays supposed to be transparent? I cannot see the data points, tick marks, or the mixing lines described in the legend. The y axis is labeled incorrectly. Inconsistent use of () in the axes labels compared to Figure 2 and 3. - corrected

Figure 2: The x-axis continues past data points. Either adjust the x-axis so it stops at 30Ma or continue the benthic foram record. It might be useful to indicate the geologic periods/the EOT on here explicitly. –the x-axes were adjusted, the proper scale for the $\delta^{18}\text{O}_{\text{benthic}}$ added, the EOT indication in the figure also inserted.

Figure 3: The axes labels are floating in space. The caption has grammatical and typographic errors. - corrected

How is are the late W-R samples identified? It looks like an arbitrary rectangle in this figure, and this is not explained in the text. - The box was removed and an arrow indicating the possible δD shift related to the interaction with late hydrothermal fluids was inserted. The latter has been inferred on the basis of anomalous δD values in biotites of samples collected in areas of the intrusion characterized by the occurrence of late small dykes were that may have disturbed the main hydrothermal assemblage. This is also explained in the text [lines: 167-170].

Reviewer #3 (Remarks to the Author):

This manuscript presents $\delta^{18}\text{O}$ and δD values of meteoric waters for the late Eocene to early Oligocene of Antarctica derived from the geochemistry of crystalline rocks. The manuscript is presents interesting data and ideas and is broadly well written, albeit with some instances where minor clarification is required. Some minor clarification are also required for the figures. I am not an expert in crystalline rock geochemistry so an unable to comment in an informed manner on the systematics or methods of this, so have focused on the interpretation and paleo-environmental discussion. The key result of a marked shift in the hydrology of the continent at the EOT is an interesting and important finding, but it is important to see these results better integrated and compared to previous studies which have looked at this using plant wax δD and other paleo-environmental information. See more specific comments below.

I recommend including line numbers in future submissions for ease of review. - line numbers inserted.

Introduction: towards the end of the first paragraph, 'affecting continental and oceanic water balance'. Can you clarify what you are meaning by this phrase. – [line 48-49]: we inserted "(ground water availability and evaporation)" to clarify the meaning of water balance

'SST were still high at the Eocene-Oligocene transition'. The references included here primarily refer to CO₂ trends through this interval, but there is evidence for cool Antarctic-proximal temperatures by the late Eocene i.e see papers mentioned below.

I suggest adding a comment into the sample description on how these samples were dated. Even though this in the supplementary, I'd recommend a comment up front especially as you refer to the error on these age estimates later in the text. – [line 71] We quoted the reference papers (Rocchi et al., 2002, Dallai and Burgess, 2011) reporting Ar-dating used to define the ages of the studied rocks.

Last paragraph of 'Interaction of meteoric waters with intrusive rocks'. Second sentence- when you refer to 'minimum temperatures at 34 Ma' from your data, what temperature is this referring too? Air? You explain later in the paragraph how you calculate temperature from your data, so I'd recommend some rearranging here to improve clarity. – [line 151-154]: we specified that δ/T rates are referred to surface temperatures

Second sentence in that paragraph. I recommend a more recent reference here, as a lot of proximal proxy temperature records have been published around the continent in the last decade i.e. as summarised in Tibbett et al. 2023, *Paleoceanography and Paleoclimatology*, and a recent Ross Sea record close to the sites in this study in Duncan et al., 2022, *Nature Geoscience*. Tibbett et al. (2023) is a useful comparison for your numbers here, as their compiled SST proxies cool by up to 3°C and MAT by up to 4°C between the late Eocene and early Oligocene, and so support your statement that using modern δ/T °C rates reconstructs a temperature difference greater than seen in the proxy records. However, I would alter the wording in your abstract where you state that your results find higher temperature variations than recorded by ocean proxies, as you discuss later that this is likely due to the modern relationship being unsuitable to use as an inference for absolute temperature change. A 19°C difference is certainly much larger than would be supported by a wide range of both proxy SST and MATs as well as other paleo-environmental information.

In the abstract of the paper we removed the sentence referring to inconsistent temperature variations. These are discussed later in the text. Specifically, [lines 143-147] we referred to the compiled MAT proxies reported in Tibbett et al. 2023 and Hutcinson et al., 2021.

There are several key references missing here that would inform and largely support your findings based on δD records preserved in plant waxes (a Miocene discussion in Feakins et al., 2012 *Nature Geoscience*, and EOT discussions in Feakins et al., 2014, *EPSL* and Tibbett et al., 2021, *Paleoceanography and Paleoclimatology*). In particular Feakins et al., 2014 use isotope enabled modelling to look at precipitation over the EOT, and forms an important comparison for this work.

[line 191]: The estimated δD values for waters in the Antarctic Peninsula at 35 Ma (Feakins et al, 2014) were introduced to possibly constrain the δD values calculated in this work.

I recommend having a location figure in the main text.

Figure 1b: what do the two different colour sample points represent? The two colors in forme Fig. 1 represented intruded granites and the Cenozoic monzonite, respectively. In the new Fig. 1b this differentiation is avoided because it does not imply any difference in the reconstruction of meteoric

water $\delta^{18}\text{O}$ composition. The petrographic characteristics of investigated lithotypes from the Mt Mc Gee outcrop are specified in the Supplementary files, as well as those of the other outcrops.

Figure 2: If the blue record is the benthic foraminifera record from Westerhold et al., 2020. Therefore it should have a secondary axis to show the actual values.- The second axis was introduced in the figures; now the samples are not scaled anymore, but actual values are reported.

Figure 3: Add the definition of WR to figure caption. - WR definition has been inserted in the caption.

Reference 23 needs a title - corrected

Supplementary:

Fig1.S1: Please add a box on the inset to show what part of Antarctica the main map represents. Does the grey just represent exposed rock rather than ice? Or a specific type of bedrock?

The grey represent exposed rock. A tiny red inset representing the study area in Antarctica has been inserted in Supplementary Fig. 1

REVIEWER COMMENTS

Reviewer #1 (Remarks to the Author):

I am satisfied with the modifications of the paper. The authors have attended to all of the posts raised. Especially Fig 1 is now more intuitive to the reader. I suggest publication of the manuscript without further changes. Looking forward to see this in print.

Reviewer #2 (Remarks to the Author):

This a review of the first revision of Dallai and Sharp.

The results will be of interest to the paleoclimate community. This is a relatively novel technique to estimate the isotopic composition of meteoric waters during Cenozoic paleoclimate transitions. There are previous studies using W/R ratios and triple oxygen isotopes to determine meteoric water compositions (Herwatz, Bindeman, and Chamberlain research groups). The novelty here is an Antarctic sample set that spans the EOT.

The figures are much improved over the previous version. Now I can see the points. Based on the other reviewers' comments, it is possible that the version that I downloaded were illegible because of software compatibility issues.

I also found the text was improved compared to the first version.

The manuscript needs improved clarity on how you calculated $d_{18}O_w$ and associated errors. From the main text and the supplement, it is still not clear you have calculated the final $d_{18}O$ value for meteoric-hydrothermal waters. I have the following questions:

- o on line 121, you state that you are assuming a $d_{18}O_{water}$ value. I thought this is what you were solving for? The 'Model Data' table in the supplement only has the final calculated $d_{18}O_{water}$ value for each point, not a range of values.
- o Gray dashed lines in Figure 1: what are these lines? How are they different from the W/R lines? As I said above - the supplement only gives a single $d_{18}O_{water}$ value for each outcrop. It does not explain what these lines are. It is particularly confusing that in some panels the gray lines intersect the large black dot that represents the final $d_{18}O_w$ value, and in other panels it does not intersect. Please report an equation that solves for $d_{18}O_w$.
- o How are you calculating what curve is the best fit for your data? I see Reviewer #1 had the same question, but I did not find the response satisfactory.
- o How are you calculating the error on the $d_{18}O_{water}$? This is not explained in the main text or in the supplement. I see Reviewer #1 had the same question.

Line 194 and elsewhere: I would argue that most of the samples from 34Ma and 35Ma do not intersect the GMWL. For these time periods, only one of several samples intersects the GMWL. This does not support the idea that the "GMWL holds for the geologic past" (Line 196). Can you explain why most of the 34Ma and 35Ma samples do not fall on the GMWL? I made a similar comment in the first submission.

Finally, there remain numerous grammatical errors throughout the text. Here's an incomplete catalog:

- line 23 -26 run on sentence
- line 33: reach  reaching
- line 77: composition is OR compositions are (subject-verb agreement)
- line 108: "in order our mixing curves to have" missing a word in here
- line 117: waters should be possisive, or say the oxygen isotope composotion of meteoric waters

- line 166: extra comma OR change 'that' to 'which'
- line 170: extra comma, subject-verb agreement is incorrect
- line 197: that to which
- line 233: extra colon
- line 240: precipitation should be singular
- line 247: could -> would

Reviewer #3 (Remarks to the Author):

I'm pleased to see that many of the comments from my first review have been addressed, and in general there is much better integration with other paleo records. There are still several points that need some minor clarification in the text. Please see below for more specific suggestions. In general, the text could do with some grammatical checks and editing for clarity.

Lines 49-53: This section still needs some clarification. The first sentence states that CO₂ declined but SSTs remained high, but then the second sentence suggests that declining CO₂ led to cooling and drying, and as I mentioned in my original review there are records of cooling circum-Antarctic SSTs in the late Eocene. Please reword these sentences to make the meaning clearer.

Line 129: 'curve' not 'curved'

Line 139: Do you mean there is 10 degrees of cooling in the Liu 2009 paper? I suggest you rearrange this paragraph to put the more recent data model comparison papers you discuss at the end of the paragraph to the start, as these are the most recent assessments of compiled high latitude data.

Line 158: This suggestion needs to be explained more, it appears as you state above that your data shows that the modern $\delta/T^{\circ}\text{C}$ relationship is unsuitable to use to derive absolute temperature change. I don't then think this necessary means Antarctica wasn't 'isolated' until after this point, as the meaning of 'isolation' can be quite subjective. This $\delta/T^{\circ}\text{C}$ relationship is just occurring against quite a different background state whereby the ice sheets exist in a climate that is still quite a lot warmer and wetter than now even though it is cooler and dryer than earlier in the Eocene (i.e. vegetation is still able to subsist on the continent). Modern-like hyper-arid polar conditions likely didn't develop until much later in the Cenozoic.

Line 224: Suggest referring to this as a 'proto-Antarctic circumpolar current' as a 'modern-like' deep and vigorous circumpolar current likely didn't develop until the late Miocene (i.e. see new paper Evangelinos et al., 2024).

Fig. 1_SI: This figure is very blurry, could the resolution be improved please.

REVIEWER COMMENTS

Reviewer #1 (Remarks to the Author):

I am satisfied with the modifications of the paper. The authors have attended to all of the posts raised. Especially Fig 1 is now more intuitive to the reader. I suggest publication of the manuscript without further changes. Looking forward to see this in print.

Reviewer #2 (Remarks to the Author):

This a review of the first revision of Dallai and Sharp.

The results will be of interest to the paleoclimate community. This is a relatively novel technique to estimate the isotopic composition of meteoric waters during Cenozoic paleoclimate transitions. There are previous studies using W/R ratios and triple oxygen isotopes to determine meteoric water compositions (Herwatz, Bindeman, and Chamberlain research groups). The novelty here is an Antarctic sample set that spans the EOT.

The figures are much improved over the previous version. Now I can see the points. Based on the other reviewers' comments, it is possible that the version that I downloaded were illegible because of software compatibility issues.

I also found the text was improved compared to the first version.

The manuscript needs improved clarity on how you calculated d18O_w and associated errors. From the main text and the supplement, it is still not clear you have calculated the final d18O value for meteoric-hydrothermal waters. I have the following questions:

1_ on line 121, you state that you are assuming a d18O_w value. I thought this is what you were solving for? The 'Model Data' table in the supplement only has the final calculated d18O_w value for each point, not a range of values. - [Supplementary Table 3_Model Data] "Model Data" report the $\delta^{18}\text{O}_w$ values used to calculate the curves fitting through the data points. The mass balance equation is solved for $\delta^{17}\text{O}_{qz}^f$ and $\delta^{18}\text{O}_{qz}^f$ at different W/R ratios.

2_ Gray dashed lines in Figure 1: what are these lines? How are they different from the W/R lines? As I said above - the supplement only gives a single d18O_w value for each outcrop. It does not explain what these lines are. It is particularly confusing that in some panels the gray lines intersect the large black dot that represents the final d18O_w value, and in other panels it does not intersect. Please report an equation that solves for d18O_w. - [Supplementary Table 3 and caption of Fig. 1a_e] The Model data report the Range of values for which the curved calculated from the mass balance equation fit through the data. The grey lines are W/R lines calculated for the $\delta^{18}\text{O}_w$ values of the range. We added this specification in the caption of Fig. 1a_e.

3_ How are you calculating what curve is the best fit for your data? I see Reviewer #1 had the same question, but I did not find the response satisfactory. - [Supplementary Table 3] The mass balance equation we used to calculate the curves is derived from that of Taylor, 1977 (Taylor, H. P. Jr. Water/rock interactions and the origin of H₂O in granitic batholiths. Journal of the Geol. Soc. Lond. 33, 509-558 (1977)) using thermometric parameters from Sharp et al., 2016 (Sharp, Z. D., Gibbons, J. A., Maltsev, O., Atudorei, V., Pack, A., Sengupta, S., Shock, E. L., Knauth, L. P. A

calibration of the triple oxygen isotope fractionation in the SiO₂-H₂O system and applications to natural samples. *Geochim. Cosmochim. Acta* 186, 105–119 (2016)).

4_ How are you calculating the error on the d18O_{water}? This is not explained in the main text or in the supplement. I see Reviewer #1 had the same question. – [Supplementary Table 3_ *Model Data*] The “±” refers to the range of the δ¹⁸O_w values from the mean value that is reported as the main curve of Fig. 1a_e. It is not the error but the possible variability of the δ¹⁸O_w values that satisfy the data. We specified “Range of δ¹⁸O_w values” fitting through the data in the table column.

5_Line 194 and elsewhere: I would argue that most of the samples from 34Ma and 35Ma do not intersect the GMWL. For these time periods, only one of several samples intersects the GMWL. This does not support the idea that the "GMWL holds for the geologic past" (Line 196). Can you explain why most of the 34Ma and 35Ma samples do not fall on the GMWL? I made a similar comment in the first submission. - We agree partially with this comment; thus we added “possibly” holds...[line 192]. In fact, most samples plot on, or scatter close, to the GMWL. The low-δD samples from 42Ma and 38Ma outcrops are interpreted in terms of late W/R interaction [line 160-165]. Among the 32Ma samples from Cape King outcrop, one sample intercepts the GMWL and the other four do not. As noted by reviewers in the comments of the first submission, the studied outcrops are characterized by low W/R ratios; thus the range of variability, that is ± 3‰ for the younger samples, needs to be taken into account. By doing this the data are even closer to the GMWL.

Finally, there remain numerous grammatical errors throughout the text. Here's an incomplete catalog:

- line 23 -26 run on sentence - divided into two sentences;
- line 33: reach  reaching – [now line 32] corrected;
- line 77: composition is OR compositions are (subject-verb agreement) – [now lines 74-75] corrected;
- line 108: "in order our mixing curves to have" missing a word in here? - [now lines 105-106] rephrased;
- line 117: waters should be possessive, - [now line 114] changed to: “the oxygen isotope composition of meteoric waters”;
- line 166: extra comma OR change 'that' to 'which' - [now line 160] the comma was removed;
- line 170: extra comma, subject-verb agreement is incorrect – [now line 164] corrected and changed with: “, which”;
- line 197: that to which - [now line 191] corrected;
- line 233: extra colon - [now line 227] corrected;
- line 240: precipitation should be singular - [now line 234] corrected;
- line 247: could -> would - [now line 241] corrected.

Reviewer #3 (Remarks to the Author):

I'm pleased to see that many of the comments from my first review have been addressed, and in general there is much better integration with other paleo records. There are still several points that need some minor clarification in the text. Please see below for more specific suggestions. In general, the text could do with some grammatical checks and editing for clarity.

Lines 49-53: This section still needs some clarification. The first sentence states that CO₂ declined but SSTs remained high, but then the second sentence suggests that declining CO₂ led to cooling

and drying, and as I mentioned in my original review there are records of cooling circum-Antarctic SSTs in the late Eocene. Please reword these sentences to make the meaning clearer. - [now lines 48-50] the sentence was re-worded as: "Climate models indicate that the effects of declining atmospheric $p\text{CO}_2$ concentration grew progressively, and co-vary with the records of cooling circum-Antarctic SSTs in the late Eocene" and the Duncan et al. 2022 reference was inserted.

Line 129: 'curve' not 'curved' - [now line 126] corrected;

Line 139: Do you mean there is 10 degrees of cooling in the Liu 2009 paper? I suggest you rearrange this paragraph to put the more recent data model comparison papers you discuss at the end of the paragraph to the start, as these are the most recent assessments of compiled high latitude data. -[now lines 135-142] we moved the more recent MAT data at the start of the paragraph and re-numbered the references.

Line 158: This suggestion needs to be explained more, it appears as you state above that your data shows that the modern $\delta/T^\circ\text{C}$ relationship is unsuitable to use to derive absolute temperature change. I don't then think this necessary means Antarctica wasn't 'isolated' until after this point, as the meaning of 'isolation' can be quite subjective. This $\delta/T^\circ\text{C}$ relationship is just occurring against quite a different background state whereby the ice sheets exist in a climate that is still quite a lot warmer and wetter than now even though it is cooler and dryer than earlier in the Eocene (i.e. vegetation is still able to subsist on the continent). Modern-like hyper-arid polar conditions likely didn't develop until much later in the Cenozoic. - [now line 152] we agree with this observation. According to present data the discussion on "Antarctica isolation" may be just speculative. For this reason we removed the last part of the sentence.

Line 224: Suggest referring to this as a 'proto-Antarctic circumpolar current' as a 'modern-like' deep and vigorous circumpolar current likely didn't develop until the late Miocene (i.e. see new paper Evangelinos et al., 2024). - [now line 218] A "proto" has been added to the ACC and the Evangelinos et al., 2024 reference has been inserted.

Fig. 1_SI: This figure is very blurry, could the resolution be improved please. [In SI] The SI_Fig.1 has been re-drawn and changed. The resolution should now be ok.

REVIEWERS' COMMENTS

Reviewer #2 (Remarks to the Author):

My comments have been mainly addressed.

We will have to agree to disagree re: Fig 3. Most samples do not fall on the GMWL because they are affected by W/R interactions, as you describe. The W/R trajectories intersect the GMWL (mostly). You are saying that the samples that fall on the GMWL support the modern GMWL, which is a bit circular in my opinion, but ok. The samples that fall off the GMWL can be explained with W/R interactions. I'll point out that even amongst the endmembers close to the GMWL, their variation around the GMWL far exceeds that of modern Antarctic precipitation shown in your figure.

Comment on Fig 1 caption:

"The dotted lines define $\delta^{18}\text{O}$ meteoric waters values, which fit through the measured data for the range of values reported in the model data (SI_Tab 3/Model Data)."

In your response to review, you say:

"The grey lines are W/R lines calculated for the $\delta^{18}\text{O}$ values of the range"

These two sentences have slightly different meanings to me (in one, you are plotting water values, in the other, mineral values).

If I may, I suggest: "The gray, dotted lines are also W/R mixing curves (same as the black lines), but they are the W/R curves that best fit the varied model parameters of WHAT values (Table S3). Their end defines the infinite water value. The best fit W/R mixing line defines the $\delta^{18}\text{O}$ of Eocene waters."

This adds specificity and should reduce confusion.

We changed Fig. 1 caption according to reviewer#2's suggestions and inserted – [lines 473-477]: The gray, dotted lines are also W/R mixing curves (same as the black lines), but they are the W/R curves that fit the varied model parameters reported in the model data (SI_Model Data). Their end also define s the infinite water value. The best fit W/R mixing line defines the $\delta^{18}\text{O}$ of Eocene waters.

Reviewer #3 (Remarks to the Author):

I am satisfied that the authors have adequately addressed my concerns from the previous review and am happy to see the manuscript published.